# A vaccine using *Anaplasma marginale* subdominant type IV secretion system recombinant proteins was not protective against a virulent challenge

**Macarena Sarli**[1,2]*, **María B. Novoa**[1,2], **Matilde N. Mazzucco**[1], **Marcelo L. Signorini**[1,2], **Ignacio E. Echaide**[1], **Susana T. de Echaide**[1], **María E. Primo**[1,2]

1 Instituto Nacional de Tecnología Agropecuaria, Estación Experimental Agropecuaria Rafaela, Rafaela, Santa Fe, Argentina, 2 Consejo Nacional de Investigaciones Científicas y Técnicas (CONICET), Rafaela, Santa Fe, Argentina

* sarli.macarena@inta.gob.ar

**Data Availability Statement:** All relevant data are within the paper and its Supporting Information files.

## Abstract

*Anaplasma marginale* is the most prevalent tick-borne livestock pathogen with worldwide distribution. Bovine anaplasmosis is a significant threat to cattle industry. Anaplasmosis outbreaks in endemic areas are prevented via vaccination with live *A. centrale* produced in splenectomized calves. Since *A. centrale* live vaccine can carry other pathogens and cause disease in adult cattle, research efforts are directed to develop safe recombinant subunit vaccines. Previous work found that the subdominant proteins of *A. marginale* type IV secretion system (T4SS) and the subdominant elongation factor-Tu (Ef-Tu) were involved in the protective immunity against the experimental challenge in cattle immunized with the *A. marginale* outer membrane (OM). This study evaluated the immunogenicity and protection conferred by recombinant VirB9.1, VirB9.2, VirB10, VirB11, and Ef-Tu proteins cloned and expressed in *E. coli*. Twenty steers were randomly clustered into four groups (G) of five animals each. Cattle from G1 and G2 were immunized with a mixture of 50 µg of each recombinant protein with Quil A® or Montanide™ adjuvants, respectively. Cattle from G3 and G4 (controls) were immunized with Quil A and Montanide adjuvants, respectively. Cattle received four immunizations at three-week intervals and were challenged with $10^7$ *A. marginale*-parasitized erythrocytes 42 days after the fourth immunization. After challenge, all cattle showed clinical signs, with a significant drop of packed cell volume and a significant increase of parasitized erythrocytes (p<0.05), requiring treatment with oxytetracycline to prevent death. The levels of IgG2 induced in the immunized groups did not correlate with the observed lack of protection. Additional strategies are required to evaluate the role of these proteins and their potential utility in the development of effective vaccines.

## Introduction

Bovine anaplasmosis is an infectious disease caused by the obligate intraerythrocytic Gram-negative bacterium *Anaplasma marginale* (order Rickettsiales; family Anaplasmataceae) [1],

**Funding:** This work was supported by ANPCyT (PICT 2013-0369) and TCP INTA EEA Rafaela. Asoc. Coop. 426100. The funders had no role in study design, data collection and analysis, decision to publish, or preparation of the manuscript.

**Competing interests:** The authors have declared that no competing interests exist.

transmitted either biologically by ticks or mechanically by bloodsucking flies or through blood-contaminated fomites. The disease is widely distributed in tropical and subtropical regions of the world, and is in expansion due to the movement of cattle from endemic to non-endemic areas [2,3]. Anaplasmosis, clinically characterized by anemia, hyperthermia, icterus, weight loss, and reduced milk production, can produce 50% mortality in cattle older than 2 years of age that have not received specific treatment [1,4]. Cattle that overcome the acute infection remain persistently infected for life and become a reservoir for *A. marginale* transmission [1,5].

In some countries, the disease is currently prevented by the administration of a live vaccine, based on the naturally less pathogenic *A. marginale* subsp. *centrale* (hereafter *A. centrale*), amplified in splenectomized calves [1]. Some drawbacks of the live *A. centrale* vaccine include the risk of transmission of other pathogens [6], the administration only to calves up to 10 months of age, and the achievement of partial protection against antigenically diverse *A. marginale* strains [1,7,8].

Immunization of cattle with the native purified outer membrane (OM) of *A. marginale* has induced complete protection against infection and clinical disease [4,9,10]. Such protection was correlated with induction of high titers of IgG2 opsonizing antibodies against *A. marginale* surface epitopes and macrophage activation mediated by CD4+ T cells [4,11]. The capacity of OM native proteins to induce protection has promoted their consideration as vaccine candidates [12,13]. However, this immunogen has been used only experimentally due to difficulties in scaling up and standardization [14].

Antibody response in OM-vaccinated cattle is primarily directed against several immunodominant major surface proteins (MSPs); however, these proteins failed to provide consistent and complete protective immunity when used individually [15–17]. Complete genome sequencing and proteomic studies of *A. marginale* allowed the identification of subdominant proteins, which are present in low abundance on the OM [13]. These proteins remain invariant during infection and are highly conserved among different strains, making them attractive potential candidates for vaccines [12,18]. Subdominant proteins of *A. marginale* type IV secretion system (T4SS), a 1.05-MDa complex that spans the outer and inner bacterial membranes involved in the host cell adhesion/invasion, and the subdominant elongation factor-Tu (Ef-Tu), a membrane-associated protein belonging to the family of hydrolases involved in protein synthesis, are targets for neutralizing antibodies [12,19,20]. The T4SS proteins VirB9.1, VirB9.2, VirB10, and VirB11 and the Ef-Tu have been recognized by sera from cattle immunized with OM that withstood the challenge with a virulent strain of *A. marginale* [12,21,22].

In the present study, the immune protection against *A. marginale* induced by a vaccine based on the recombinant proteins VirB9.1, VirB9.2, VirB10, VirB9.1, and Ef-Tu was evaluated in cattle.

## Material and methods

### Cattle

The cattle involved in this research were born and raised in an anaplasmosis-free Holstein dairy herd in Rafaela (31˚12'S-61˚30'W), a zone free from the cattle tick *Rhipicephalus microplus* in Argentina. The study group included a 4-month-old splenectomized calf used to amplify *A. marginale* and 20 2-year-old healthy steers used for vaccine evaluation, which were maintained in different isolation pens. All cattle received forage, concentrate and drinking water *ad libitum*. They were sprayed weekly with flumethrin (Bayticol® Pour-On, Bayer) to protect them from biting flies. All the animals were confirmed to be free of *Anaplasma* spp. infection by cELISA and nested PCR (nPCR) before the start of the experiment [23,24]. All

procedures were approved by the Animal Care Committee of the Faculty of Veterinary Sciences, National University of Litoral (Protocol number 243/15).

## Genomic DNA

DNA was purified from 900 μL of *A. marginale*-infected blood [25]. Briefly, erythrocytes were lysed with erythrocyte lysis buffer (0.14 M $NH_4Cl$, 0.17 M Tris—HCl) and the hemoglobin was eliminated by washes with distilled water. The inclusion bodies were lysed in 400 μL of cellular lysis buffer (0.05 M Tris—HCl, 0.1 M EDTA, 0.1 M NaCl, 2% SDS, pH 8) with 160 μg of proteinase K. DNA was extracted with phenol/chloroform/isoamyl alcohol, precipitated with isopropyl alcohol and washed with 75% ice-cold ethanol. The pellet was suspended in 50 μL distilled water and kept at -20°C until use. The concentration and purity of DNA was assessed at 260–280 nm (NanoDrop 2000, Thermo Fisher Scientific, USA).

## In silico analysis of proteins sequences

The prediction algorithm SignaIP 5.0 (http://www.cbs.dtu.dk/services/SignalP/) was used to predict signal peptides [26]. TMpred algorithm was used to predict the transmembrane domains of each protein (http://embnet.vital-it.ch/software/TMPRED_form.html) [27]. Solubility of the full-length and truncated form (without transmembrane domains) of the proteins was calculated using a prediction model based on overexpression of heterologous proteins in *E. coli* (http://www.biotech.ou.edu/) [28].

## Cloning of DNA sequences

The recombinant proteins VirB9.1 and VirB9.2 were cloned and expressed as truncated form, without the signal peptide (tVirB9.1 and tVirB9.2). cDNA encoding residues 22–272 of VirB9.1 (AAV86251.1), 27–281 of VirB9.2 (AAV87107.1) and full-length sequences of VirB10 (AAV87106.1), VirB11(RCL20095.1), and Ef-Tu (WP_037348707.1) proteins were amplified by PCR using specific primers designed *ad hoc* (Table 1). The sequence for a six histidine tag (6 His-tag) at the C-terminus of tVirB9.2, VirB10, VirB11, and Ef-Tu proteins was added to the reverse primers (Table 1). Amplicons were cloned into pGEM-T Easy vector (Promega, USA) according to the manufacturer's instructions. *E. coli* JM 109 competent cells (Promega) were transformed with the recombinant plasmids. Subsequently, a fragment was excised with restriction enzymes *Nde*I and *Bam*HI (sites shown in Table 1), and subcloned into pET9b (Novagen, USA) to yield the ptVirB9.1, ptVirB9.2, pVirB10, pVirB11, and pEf-Tu plasmids. The identity of the DNA constructs was confirmed by sequencing (Biotechnology Institute, INTA CICVyA, Argentina). The constructs were used to transform *E. coli* BL21 RIL (DE3) *pLysS* competent cells (Novagen).

**Table 1. Primers designed *ad hoc* for amplifying the DNA sequences of tVirB9.1, tVirB9.2, VirB10, VirB11, and Ef-Tu proteins from *Anaplasma marginale*.**

| Protein name | Forward primer (5′ to 3′) | Reverse primer (5′ to 3′) |
|---|---|---|
| tVirB9.1 | catatgcaggaaccgcgctctatag | ggatcctttaaacccacgtccccttctggatg |
| tVirB9.2 | catatggtaagcggtggtg | ggatcctcagtgatggtgatggtgatggcggccttcaattttaaaaagcaccg |
| VirB11 | catatgacagcaggatacgcagcgttag | ggatccctagtgatggtgatggtgatgttaaaatcattgccttgtgaacatttagtg |
| VirB10 | catatgtcagacgaaaccaaggataataac | ggatccctagtgatggtgatggtgatgtttaaacctacgcaccgcctccc |
| Ef-Tu | catatgacagaagggagaaagcc | ggatccctagtgatggtgatggtgatgtttaaactccaaaatctcagttatg |

Sites of the restriction enzymes *Nde*I and *Bam*HI are underlined. The sequences that codify for the six-histidine tag are shown in italics.

## Protein expression and purification

*E. coli* BL21 RIL (DE3) *pLysS* competent cells carrying the plasmids ptVirB9.1, ptVirB9.2, pVirB10, pVirB11, or pEf-Tu were cultured at 37 ˚C in 500 mL of Luria–Bertani medium, supplemented with 50 μg/mL kanamycin and 34 μg/mL chloramphenicol to $OD_{600nm}$ = 1. Protein expression was induced with 1% lactose for 3 h; then, *E. coli* cells were harvested by centrifugation. The cells expressing tVirB9.2, VirB10, VirB11, and Ef-Tu proteins were suspended in 10 mL of lysis buffer A (50 mM sodium phosphate, 300 mM NaCl, 10 mM imidazole, pH 8) and those expressing tVirB9.1 were suspended in 10 mL of lysis buffer B (50 mM sodium phosphate, pH 7.2), both containing 1:1000 protease inhibitor cocktail set III (Calbiochem, USA). Finally, the cells were lysed twice at 20,000 psi using a cell disruptor (Avestin Emulsiflex B15, Canada). Soluble and insoluble fractions were isolated by centrifugation (12,000 x*g*, 4˚C, 30 min).

The tVirB9.1 and Ef-Tu proteins were purified from the soluble fraction of the *E. coli* lysate. The soluble fraction of tVirB9.1 was precipitated by adding saturated ammonium sulfate up to 30% saturation with stirring for 20 min on ice. The precipitated protein was separated from the supernatant by centrifugation (12,000 x*g*, 4 ˚C, 20 min). After desalting by dialysis in buffer C (25 mM Tris–HCl, pH 7.2), tVirB9.1 was purified by ionic exchange chromatography on Q-sepharose fast flow resin (GE Healthcare, USA) equilibrated with the same buffer. The resin was washed with five volumes of buffer C containing 50 mM NaCl; then, the protein was eluted with five volumes of buffer C containing 200 mM NaCl. The soluble fraction of Ef-Tu was added to 2 mL of $Ni^{+2}$-NTA agarose (Qiagen, Germany) previously equilibrated with lysis buffer A. After incubation at 4 ˚C for 1 h, the suspension was poured into a 1.5 cm x 5.0 cm column and washed with five volumes of 30 mM imidazole lysis buffer A. The protein was eluted successively with five volumes of 100 mM and then with five volumes of 200 mM imidazole lysis buffer A.

The proteins tVirB9.2, VirB10, and VirB11 were purified under denaturing conditions. The inclusion bodies were obtained from the insoluble fraction of the *E. coli* lysate and washed three times using successively 1% of triton X-100, 2% of triton X-100 and 2 M urea, each diluted in the wash buffer (50 mM Tris–HCl, 5 mM EDTA, 5 mM DTT, pH 8) and twice with ultrapure water. All washes were performed by centrifugation at 12,000 x*g*, at 20 ˚C for 15 min. The isolated inclusion bodies were solubilized by incubation at 25 ˚C for 3 h in denaturing buffer D (100 mM sodium phosphate, 8 M urea, 5 mM β-mercaptoetanol, pH 8) for VirB10 and VirB11. The denaturing buffer E (100 mM sodium phosphate, 6 M guanidinium chloride, 5 mM β-mercaptoethanol, pH 8) was used to solubilize tVirB9.2. After centrifugation (12,000 x*g*, 20 ˚C, 30 min), the soluble fraction was added to 2 mL of $Ni^{+2}$-NTA agarose previously equilibrated with the corresponding denaturing buffer. After incubation at 20 ˚C for 1 h, the suspension was poured into a column and washed with five volumes of 30 mM imidazole denaturing buffer D. The proteins were eluted with five volumes of 200 mM imidazole denaturing buffer D.

The purity of the proteins was assessed by sodium dodecyl sulphate-polyacrylamide gel electrophoresis (SDS-PAGE) using a standard protocol [29]. The molar concentration in pure samples was calculated by absorbance at 280 nm using the molar extinction coefficient ($\varepsilon_{280nm}$) of each protein (25600 $M^{-1}cm^{-1}$, 26360 $M^{-1}cm^{-1}$, 11710 $M^{-1}cm^{-1}$, 25900 $M^{-1}cm^{-1}$, and 26025 $M^{-1}cm^{-1}$ to tVirB9.1, tVirB9.2, VirB10, VirB11, and Ef-Tu, respectively).

Pure recombinant proteins were analyzed by Western blot (WB). Proteins were electrophoresed (0.2 μg/lane) on a 12% SDS-PAGE and transferred to a nitrocellulose membrane (Trans-Blot® 0.45 μm, Bio-Rad), by electroblotting at 50 V for 2 h. The membrane was blocked in TBS (50 mM Tris–HCl, 150 mM NaCl, pH 7.6)/5% nonfat dried milk overnight at 4 ˚C. After

five washes with TBST (TBS/0.05% Tween-20), it was incubated with mouse anti-His-tag MoAb (MA1-21315, Thermo Fisher Scientific), diluted 1:2000 in TBST/10% nonfat dried milk at room temperature for 1 h. After five washes with TBST, it was incubated with goat anti-mouse IgG peroxidase conjugate (#115-036-072 Jackson ImmunoResearch Inc., USA) at the same dilution for 1 h. The reaction was revealed by adding the colorimetric substrate 3,3'-dia-minobenzidine tetrahydrochloride (DAB) (Sigma-Aldrich, USA).

Three proteins, major surface protein 5 (MSP5) of *A. marginale*, surface antigen 1 (SAG1) of *Neospora caninum*, and merozoite surface antigen 2c (MSA2c) of *Babesia bovis* were expressed in *E. coli* and purified by pseudo-affinity on a $Ni^{+2}$-NTA agarose column following the protocol described for MSP5 protein [30]. These proteins share the sequence (`FKIEGRH HHHHH`) in the C-terminal extreme with four of the proteins used as immunogens, and were used for the anti-His-tag adsorption step or for the determination of the presence of these antibodies at the post-absorption step.

## Immunization

Two vaccine formulations based on a mixture of 50 μg of each pure recombinant protein (tVirB9.1, tVirB9.2, VirB10, VirB11, and Ef-Tu) diluted in PBS were adjuvanted with 1 mg/mL Quil A (Brenntag, Denmark) or 50% v/v Montanide ISA201 (Seppic Inc., France). Quil-A formulations were prepared by adding the solid compound into diluted proteins. Montanide formulations were prepared following Seppic's indications. Briefly, a stable emulsion W/O/W was prepared in a one-step process using a low shear rate and controlled temperature at 31˚C (+/-1 ˚C).

Twenty steers were randomly clustered into four groups (G) of five animals each (n = 5) and immunized with the recombinant proteins/Quil A (G1) or recombinant proteins/Montanide (G2) and PBS/Quil A (G3) or PBS/Montanide (G4), as controls. Cattle received four doses of 2 mL of the corresponding immunogen by subcutaneous (SC) injection in the neck, at three-week intervals (days 0, 21, 42, and 63). The presence of swelling at the immunization site was recorded. The cattle were bled once before immunization and then weekly during 10 weeks; the serum samples were stored at -20 ˚C until use.

## Challenge

An isolate of *A. marginale* from Salta (Argentina) that had been stored frozen was used to challenge the immunity of the cattle [31]. Cryopreserved parasitized blood was thawed and inoculated into a splenectomized calf [32] that was bled when the parasitemia reached 5%, in 5% sodium citrate as anticoagulant. Each challenge dose was adjusted to $10^7$ parasitized erythrocytes in a 2-mL final volume and inoculated SC to the cattle, on day 42 after the fourth immunization. The clinical reaction was monitored daily during 40 days, starting 10 days post-challenge (dpch), by measuring body temperature (T) in degrees Celsius (˚C), packed cell volume (PCV), and percentage of parasitized erythrocytes (PPE) through blood smears stained with Giemsa that were microscopically analyzed (1000x) [33]. Cattle were treated with 20 mg $kg^{-1}$ oxytetracycline (Terramicina® LA, Pfizer) to prevent death when the clinical parameters achieved ≤15 PCV, ≥5 PPE, or when the T was ≥41 ˚C during three consecutive days. The clinical parameters of each group were expressed as the mean of the maximum percent (%) drop of PCV, mean of the maximum PPE and mean of the cumulative T above 39.5 ˚C. Cattle were bled weekly during five weeks after challenge; the whole blood was analyzed by nPCR and the serum samples were tested by cELISA [23,24].

## Antibody response

An indirect ELISA (iELISA) was used to detect total IgG (IgGT), IgG1, and IgG2 specific against tVirB9.1, tVirB9.2, VirB10, VirB11, and Ef-Tu. To eliminate antibodies directed toward remnant *E. coli* proteins and the C-terminal His-tag, sera from immunized cattle were adsorbed with lysate from *E. coli* expressing SAG1. To evaluate the removal of anti-His-tag antibodies from immune sera, IgGT against *A. marginale* MSP5 and *Babesia bovis* MSA2c were measured. Optimal dilutions were established using checkerboard titrations with dilutions of sera, antigens and conjugates [34]. Polystyrene microplates (Thermo Fisher Scientific) were individually coated with 100 μL of each recombinant protein (5 μg/mL) in PBS (145 mM NaCl, 4.4 mM NaHPO$_4$, 18.3 mM NaH$_2$PO$_4$), and incubated overnight at 4 ˚C. The coated plates were washed with PBS three times and incubated with 300 μL of blocking buffer (PBS/10% nonfat dried milk) at 25 ˚C for 1 h. These conditions were also used for sera and conjugate incubations.

To assess IgGT, the plates were washed with PBST (PBS/0.05% Tween-20) three times and then incubated with 100 μL of serum samples diluted 1:10 in PBST/10% nonfat dried milk. After four washes with PBST, they were incubated with 100 μL of rabbit anti-bovine IgG peroxidase conjugate (A5295, Sigma-Aldrich), diluted 1:2000 in PBST/10% nonfat dried milk. After four washes as above, 100 μL of chromogenic substrate 1 mM 2,2'-Azinobis [3-ethylbenzothiazoline-6-sulfonic acid]-diammonium salt (ABTS) (Sigma-Aldrich) in 0.05 M sodium citrate pH 4.5, 0.03% v/v H$_2$O$_2$ was added. The absorbance was measured in a microplate reader at 405 nm at 25 ˚C, 15 min after the addition of chromogenic substrate. The cutoff point for each antigen was set as the OD$_{405nm}$ mean for the pre-immune sera (n = 20) + 3 standard deviation (SD) [35].

To assess IgG1 and IgG2 antibodies, the ELISA protocol described above was performed with a few modifications. Serum samples were diluted 1:20 and the mouse anti-bovine IgG1 MoAb (MCA627 Serotec™, UK) or IgG2 (B8400, Sigma-Aldrich) were added at a dilution of 1:1000. As second antibody, goat anti-mouse IgG peroxidase conjugate (Jackson, ImmunoResearch) at the same dilution was used. The IgG1/IgG2 ratio of OD$_{405nm}$ was analyzed for G1 and G2 [36]. All field serum samples and controls were assayed in duplicate.

## Specificity of antibodies

Anti tVirB9.1, tVirB9.2, VirB10, VirB11, and Ef-Tu antibodies from sera of G1 and G2 cattle were evaluated by WB 7 days after the fourth immunization, following the previously described protocol. Before the test, cattle sera were adsorbed by overnight incubation at 4 ˚C with an *E. coli* lysate expressing SAG1. To evaluate the removal of anti-His-tag antibodies from immune sera, MSP5 was included as a control in the WB.

The recognition of the proteins expressed in *A. marginale* by the cattle sera was evaluated by WB using a crude antigen of *A. marginale* purified from blood of a splenectomized calf with 80% of parasitemia. This *A. marginale* crude antigen was obtained following the protocol described for production of antigen of Card agluttination test [8]. In the WB, sera were diluted 1:100. The antigen-antibody reaction was detected using rabbit anti-bovine IgG peroxidase conjugate (A5295, Sigma-Aldrich) at a dilution of 1:1000.

## Statistical analysis

Data were analyzed using the software InfoStat (Universidad Nacional de Córdoba, Córdoba, Argentina) (http://www.infostat.com.ar). The levels of antibodies against recombinant proteins were compared between groups using a Generalized Linear Model of repeated measurements with Gamma distribution as a link function considering the frequency distribution of

the response variable. The means of the clinical parameters, cumulative T above 39.5˚C, maximum % drop of PCV, and maximum PPE between groups after the challenge were compared using ANOVA. Differences in the mean of antibodies anti-MSP5 (inhibition percentage) between immunized and control groups on different dpch were analyzed by Mann Whitney test. All statistical analyses were considered significant at $p < 0.05$.

## Results

### Sequences analysis, expression and purification of recombinant proteins

In silico analysis showed that VirB9.1 and VirB9.2 proteins contain a signal peptide with a cleavage site within the 20–21 and 25–26 residue region, respectively (S1 Table). No signal peptides were identified for VirB10, VirB11, or Ef-Tu proteins. The analysis of the primary structure of the proteins showed a transmembrane helix (TMH) in the N-terminus of VirB9.1 (residues 4–22) and VirB9.2 (residues 7–27) and two TMH in VirB10 (residues 29–47 and 339–357). TMH was not predicted in VirB11 or Ef-Tu. The predicted solubility for the proteins VirB9.1, VirB9.2, and VirB10, which contain a hydrophobic region, was 38.7%, 100%, and 0%, respectively. The solubility for VirB9.1 increased to 98.7% when it was expressed as truncated protein (residues 22–272). Removal of the hydrophobic region did not modify the predicted solubility for the VirB9.2 or VirB10 proteins. The VirB9.1 and VirB9.2 proteins were expressed without their signal peptide, whereas VirB10, VirB11, and Ef-Tu proteins were expressed with their full-length sequence. The recombinant proteins tVirB9.1 and Ef-Tu were expressed in soluble form in the cytoplasm of *E. coli* with a yield of 20 and 15 mg per liter of culture after their purification, respectively. The recombinant proteins tVirB9.2, VirB10, and VirB11 were expressed mainly in inclusion bodies and had a yield of 20, 10, and 12 mg per liter of culture, respectively, after their purification under denaturing conditions.

The approximate molecular masses for the purified tVirB9.1, tVirB9.2, VirB10, VirB11, and Ef-Tu were 28, 29, 49, 38, and 44 kDa, respectively (Fig 1A). The molecular size observed agrees with that expected for each protein. The recombinant proteins reacted with the MoAb anti-His tag (Fig 1B), except for tVirB9.1, which lacked the His-tag epitope.

### Adjuvant reaction

Cattle that received Montanide as adjuvant (G2 and G4) showed a small inflammatory reaction <1.5 cm at the immunization site (neck region), whereas no reaction was observed in those that received Quil A (G1 and G3).

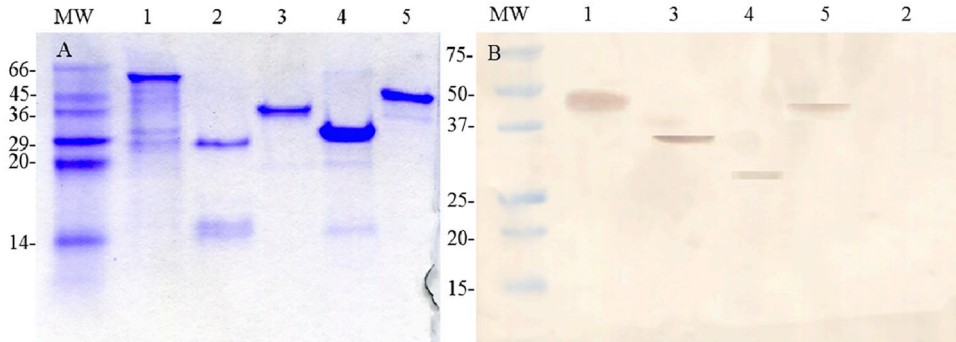

**Fig 1. Evaluation of *Anaplasma marginale* purified recombinant proteins.** (A) SDS-PAGE, stained with Coomassie Brilliant Blue R-250. (B) Western blot using anti-His-tag MoAb. MW: molecular weight marker (kDa); lane 1: VirB10; lane 2: tVirB9.1; lane 3: VirB11; lane 4: tVirB9.2; lane 5: Ef-Tu.

## Antibody response

All G1 and G2 cattle generated antibodies against each recombinant protein after immunizations, detected by iELISA, whereas G3 and G4 cattle remained negative until the challenge. The kinetics of IgGT, IgG1, and IgG2 was similar for G1 and G2, both after immunizations and after challenge with *A. marginale*. An increase of IgGT levels against tVirB9.1, tVirB9.2, VirB10, VirB11, and Ef-Tu, above the cutoff point, was detected from day 7 after the second immunization (Fig 2). From this day, G1 and G2 cattle showed similar IgGT levels (p>0.05) (Fig 2). Regarding the immune response stimulated by each antigen individually, tVirB9.1 and tVirB9.2 induced a higher level of IgGT than VirB10, VirB11, and Ef-Tu (p<0.05). At 35 dpch, G3 and G4 cattle showed a significant increment of IgGT levels (p<0.05), although the values were lower than those of G1 and G2 cattle. At this time, antibodies against *A. marginale* MSP5 protein were detected in the four groups of cattle (Fig 2).

tVirB9.1, tVirB9.2, VirB10, VirB11, and Ef-Tu but not MSP5 were recognized by antibodies present in serum samples of all G1 and G2 cattle obtained 7 days after fourth immunization, when they were evaluated by WB (Fig 3). Moreover, these antibodies recognized proteins in the *A. marginale* crude antigen (Fig 3).

The recombinant proteins tVirB9.1, tVirB9.2, VirB10, VirB11, and Ef-Tu, using Quil A or Montanide as adjuvants, induced a stronger IgG2 response than that of IgG1 both after immunizations and after challenge (p<0.05), and the IgG1/IgG2 ratio remained <1 (Table 2). All cattle from G3 and G4 showed an increase of IgG1 and IgG2 levels after challenge (p<0.05), but with lower values than those reached by G1 and G2 cattle (Fig 2).

## Response to challenge

After challenge with $10^7$ erythrocytes parasitized with *A. marginale*, cattle from all groups responded with a significant drop of PCV and a significant increase of PPE (p<0.05), requiring treatment (20 mg kg$^{-1}$ oxytetracycline) to prevent their death. After a prepatent period of 20 ± 1 days, *A. marginale* infection was confirmed through Giemsa stained blood smears in all cattle. There were no significant differences between groups in the means of the evaluated clinical parameters (p>0.05) (Table 3). The antibiotic treatment administered to cattle between days 23 and 30 after challenge attenuated the drop of PCV and the increase of PPE. The maximum % drop of PCV and the maximum PPE were recorded between days 28 and 30 after challenge in all cattle.

Infection was confirmed in all cattle by nPCR at 14 dpch. By cELISA and iELISA, all cattle were positive at 35 dpch (Fig 2). At 14 dpch, the mean of anti-MSP5 antibodies, determined by cELISA and expressed as percentage of inhibition, of the immunized groups was higher than the mean of the control groups. These results show a greater number of positive immunized animals (6/10) than that of positive controls animals (0/10).

## Discussion

In this work, five subdominant *A. marginale* proteins were tested as immunogens in cattle. These proteins were postulated as vaccine candidates, but have not been evaluated in an immunization and challenge experiment [12,22]. Recombinant tVirB9.1, tVirB9.2, VirB10, VirB11, and Ef-Tu proteins failed to induce protection against the pathogenic effects of *A. marginale* following the experimental challenge.

The outer membrane (OM) T4SS proteins are important for intracellular survival and virulence of Gram negative bacteria [12,37]. Many of those proteins are exposed on the cell surface, where they could be targeted by neutralizing antibodies. The subdominant epitopes are eligible to induce immune protection, as it was clearly established for other pathogens [38,39]. T4SS

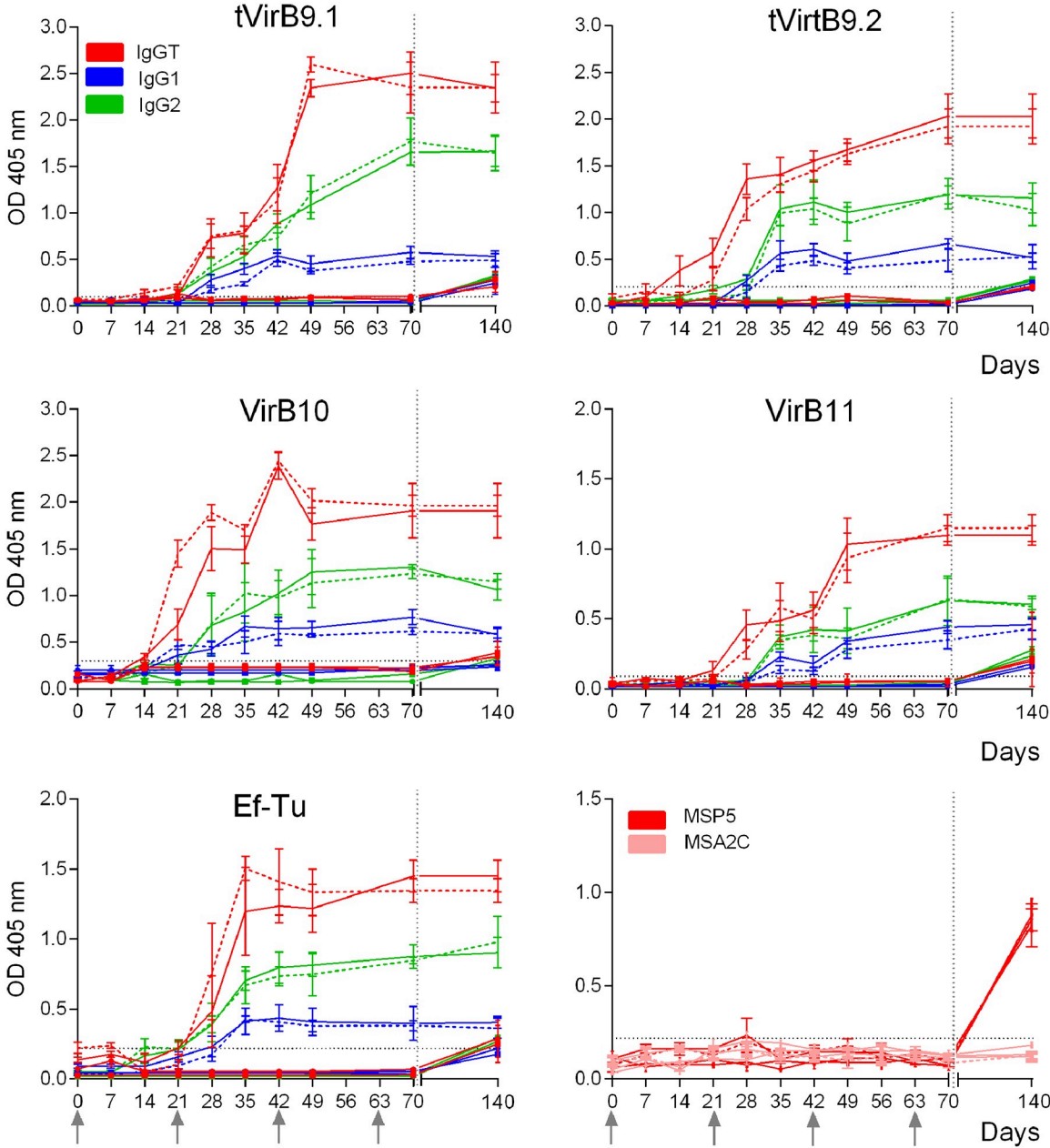

**Fig 2. Kinetics of antibody response (IgGT, IgG1, and IgG2) to each recombinant protein measured by iELISA.** Group 1, recombinant proteins/Quil A (—); Group 2, recombinant proteins/Montanide (- - -); Group 3, PBS/Quil A (—●—) and Group 4, PBS/Montanide (—■—). Each point represents the mean ± SEM of the $OD_{405nm}$ at different days after immunization and at 35 days after challenge (day 140). IgGT against *A. marginale* MSP5 and *B. bovis* MSA2c were measured as control of the presence of anti-His-tag antibodies. The horizontal dotted line indicates the cutoff point. The arrows indicate the days of the immunizations and the vertical dotted line indicates the day of challenge with *A. marginale* (on day 42 after the fourth immunization).

proteins and Ef-Tu are highly conserved among geographically distinct strains of *A. marginale* [12].

Tebele et al. (1991) and Brown et al. (1998) demonstrated that immunization of calves with fractions or the whole *A. marginale* OM, adjuvanted with saponin, induced immune protection against homologous challenge, characterized by a strong T helper cell immune response

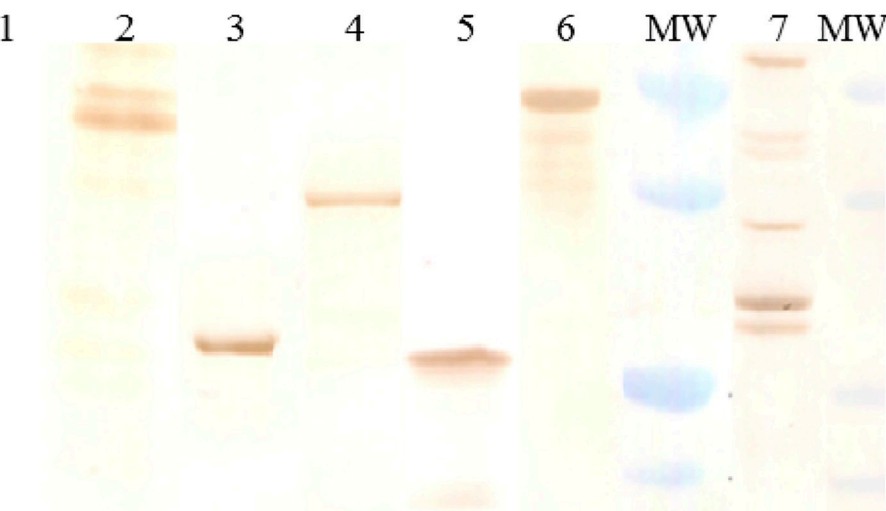

**Fig 3. Reactivity of sera obtained 7 days after the fourth immunization from cattle inoculated with recombinant proteins/Quil A (Group 1) or recombinant proteins/Montanide (Group 2) by Western blot.** A representative steer of group 1 is shown. MW: molecular weight marker (20, 25, 37, 50 kDa). Lane 1: MSP5; lane 2: Ef-Tu; lane 3: tVirB9.2; lane 4: VirB11; lane 5: tVirB9.1; lane 6: VirB10; lane 7: *A. marginale* crude antigen. Sera were diluted 1/100 and the reaction was detected with anti-bovine IgG peroxidase conjugate.

and high titers of IgG2 [4,9]. Despite the response of IgG2 observed in this work, both vaccine formulations failed to mitigate the course of infection. Other subdominant proteins from the OM, AM854 and AM936, were able to induce IgG2 immune response; however, they also failed to protect against the challenge [40]. Albarrak et al. (2012) demonstrated that the subdominant protein AM779 from the OM was unable to protect calves after the homologous challenge with adult males of *Dermacentor andersoni* infected with the *A. marginale* St. Maries strain [41]. In those experiments, IgG2 titers to subdominant proteins were similar in cattle immunized with recombinant proteins or purified OM. This finding supports the theory that antigen amount is not a primary determinant of subdominance for B cell responses [41]. The IgG2 titers specific for subdominant proteins obtained from cattle vaccinated with OM ranged from 100 to 5,000 [12,40,41], in contrast with IgG2 titers to MSP2, which were higher than 30,000 [12,41]. It is possible that the lack of protection in our work was due to low IgG2 levels; however, it has been shown that IgG2 titers do not correlate with protection [41]. In addition, the clinical signs observed in this work were similar to those reported for cattle immunized with subdominant recombinant proteins [40,41].

The antibody response to MSP5 14 dpch was observed in 60% of immunized cattle (G1 and G2), whereas control cattle (G3 and G4) remained negative. This difference could be attributed to the presence of opsonizing antibodies generated after immunization with the recombinant

**Table 2. Mean of IgG1/IgG2 ratio (OD$_{405nm}$) in the immunized groups recorded on day 7 after the fourth immunization with recombinant proteins/Quil A (Group 1) or recombinant proteins/Montanide (Group 2) and 35 days after challenge with *Anaplasma marginale*.**

| Groups | Mean of IgG1/IgG2 ratio (Day 7 after the fourth immunization/Day 35 after-challenge) | | | | |
| --- | --- | --- | --- | --- | --- |
| | **tVirB9.1** | **tVirB9.2** | **VirB10** | **VirB11** | **Ef-Tu** |
| Group 1 | 0.43/0.52 | 0.23/0.35* | 0.25/0.40* | 0.38/0.51* | 0.43/0.56 |
| Group 2 | 0.63/0.66 | 0.42/0.47 | 0.23/0.34 | 0.32/0.42 | 0.63/0.58 |

Significant differences ($p<0.05$) between pairs of values are indicated with an *

**Table 3. Mean values (± SD) of clinical parameters for each group of cattle immunized with four doses of recombinant proteins/Quil A (Group 1), recombinant proteins/Montanide (Group 2), PBS/Quil A (Group 3), or PBS/Montanide (Group 4) after challenge with $10^7$ *Anaplasma marginale* parasitized erythrocytes.**

| Groups | Maximum percent drop of PCV | Maximum PPE | Cumulative T above 39.5 ˚C | OTC-treated cattle (n/n) |
|---|---|---|---|---|
| Group 1 | 48.4 ± 5.1 | 5.8 ± 0.8 | 1.2 ± 0.5 | 5/5 |
| Group 2 | 44.4 ± 3.9 | 6.0 ± 1.0 | 1.4 ± 0.8 | 5/5 |
| Group 3 | 45.7 ± 7.7 | 6.9 ± 0.2 | 1.1 ± 0.6 | 5/5 |
| Group 4 | 45.7 ± 7.9 | 7.0 ± 0.6 | 1.4 ± 0.8 | 5/5 |

PCV, packed cell volume; PPE, percentage of parasitized erythrocytes; T, body temperature; OTC, oxytetracycline.

Values were not significantly different among groups (p>0.05)

proteins. It is well known that antibodies are better opsonins for the adaptive immune system than the complement factors of the innate immune system. Thus, after the challenge, in the immunized cattle the antibodies directed against *A. marginale* surface exposed proteins (VirB9.1, VirB9.2, and VirB10), opsonized the bacteria and were recognized, through the Fc region of the IgG, by the phagocytic cells that processed the opsonized bacteria and presented to T cells. This process would favor the early secretion of antibodies in the immunized animals.

The lack of protective immunity observed in this study could be attributed to the failure of the recombinant proteins to expose their critical epitopes with the correct conformation and generate a protective immune response. The proper conformational structure of epitopes is obtained when the recombinant proteins are expressed in native form [14,42], a task that is difficult to perform with the OM proteins of *A. marginale* due to their intrinsic characteristics. In previous works, recombinant VirB9.1, VirB9.2, VirB10, VirB11, and Ef-Tu proteins were obtained under denaturing conditions [12,21,22]. Zhao et al. (2016) were able to express VirB9.1 in the *E. coli* soluble fraction as GST-VirB9.1, whereas VirB9.2 expressed in the insoluble fraction as SUMO-VirB9.2 was then refolded [14]. The cloning and expression of heterologous recombinant proteins in *E. coli* that lack the signal peptide can increase the expression levels and the solubility of the proteins without affecting their immunogenicity [43]. In this work, in silico analysis showed that the expression of soluble VirB9.1 was improved without inclusion of the signal peptide; thus, tVirB9.1 (residues 22–272) was expressed in the *E. coli* soluble fraction. Contrary to results reported by Zhao et al. (2016) [14], refolding of tVirB9.2 without its signal peptide (residues 27–281), obtained in the insoluble fraction in *E. coli*, was not achieved. This difference could be explained by the use of the SUMO fusion protein, which prevents aggregation of folding intermediates, keeping them in solution long enough to adopt correct conformations [44].

Another cause of vaccine failure in this study may have been the inability of the immunogen to generate antibodies that block the parasite entry to the host cell. The most successful vaccines target highly conserved epitopes required by the pathogenic parasites for their host cell entry [45]. During *A. marginale* multiplication, new antigenic variants of MSP2 are generated. Studies have demonstrated that this antigenic variation of MSP2 also occurs during persistent *A. centrale* infections [46]. The conservation of CD4+ T-cell epitopes between *A. marginale*-MSP2 and *A. centrale*-MSP2, and the generation of new antigenic variants during the *Anaplasma* life cycle may contribute to the cross-protection produced by *A. centrale* live vaccine. In addition, studies have shown that multi-antigen vaccines may be more effective to induce a protective response than an individual antigen [47]. In this work, the five proteins evaluated may have been insufficient to generate antibodies capable to block the erythrocytes invasion by *A. marginale*.

The development of an effective recombinant vaccine against *A. marginale* based on subdominant antigens of the OM to block the host cells-parasite interplay requires further studies to identify the critical epitopes of these antigens, express them as native proteins and determine the protein-protein interactions.

## Supporting information

**S1 Raw images.**
(PDF)

**S1 Table. In silico analysis of protein sequences.** The signal peptide of VirB9.1 and VirB9.2, excluded from tVirB9.1 and tVirB9.2, are highlighted with horizontal gray bars. The transmembrane helices from VirB10 are indicated in italics.
(PDF)

## Author Contributions

**Conceptualization:** Macarena Sarli, Ignacio E. Echaide, María E. Primo.

**Formal analysis:** Macarena Sarli, Marcelo L. Signorini.

**Funding acquisition:** Macarena Sarli.

**Investigation:** Macarena Sarli.

**Methodology:** Macarena Sarli, María B. Novoa, Matilde N. Mazzucco.

**Supervision:** María E. Primo.

**Writing – original draft:** Macarena Sarli.

**Writing – review & editing:** Ignacio E. Echaide, Susana T. de Echaide, María E. Primo.

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
