## [Decision Letter · Decision Letter 0]

6 Nov 2019

PONE-D-19-26977

A vaccine using Anaplasma marginale subdominant type IV secretion system recombinant proteins was not protective against a virulent challenge

PLOS ONE

Dear miss Sarli,

Thank you for submitting your manuscript to PLOS ONE. After careful consideration, we feel that it has merit but does not fully meet PLOS ONE’s publication criteria as it currently stands. Therefore, we invite you to submit a revised version of the manuscript that addresses the points raised during the review process.

1) It is important to assure that the antibodies is against the antigens and not to the his-tag. A non-related protein with the same his-tag can be used as control;

2) Please, answer to all the comments raised by both reviewers.

We would appreciate receiving your revised manuscript by Dec 21 2019 11:59PM. To enhance the reproducibility of your results, we recommend that if applicable you deposit your laboratory protocols in protocols.io, where a protocol can be assigned its own identifier (DOI) such that it can be cited independently in the future. For instructions see: http://journals.plos.org/plosone/s/submission-guidelines#loc-laboratory-protocols

We look forward to receiving your revised manuscript.

Kind regards,

Paulo Lee Ho, Ph.D.

Academic Editor

PLOS ONE

Journal Requirements:

1. To comply with PLOS ONE submission requirements, in your Methods section, please ensure you have included details on (1) methods of sacrifice, (2) methods of anesthesia and/or analgesia, and (3) efforts to alleviate suffering.

Reviewers' comments:

Reviewer's Responses to Questions

**Comments to the Author**

1. Is the manuscript technically sound, and do the data support the conclusions?

Reviewer #1: Partly

Reviewer #2: Yes

2. Has the statistical analysis been performed appropriately and rigorously? 

Reviewer #1: Yes

Reviewer #2: Yes

3. Have the authors made all data underlying the findings in their manuscript fully available?

Reviewer #1: Yes

Reviewer #2: Yes

4. Is the manuscript presented in an intelligible fashion and written in standard English?

Reviewer #1: Yes

Reviewer #2: Yes

5. Review Comments to the Author

Reviewer #1: Dear Authors,

This clearly written manuscript presents negative data regarding an immunization and challenge using primarily the type 4 secretion system components of Anaplasma marginale. Overall this is nicely done work, though a few things require strengthening or clarification.

The authors need to more clearly demonstrate that they are measuring the immune response directed against the recombinant proteins and not the His tag following immunization. Though it is mentioned on line 227 that the sera were adsorbed with E. coli lysate, it is not clear that the expressed His tag was included in the E. coli lysate. Additionally, there are no controls to demonstrate that the adsorption was effective in removing the anti-his tag antibodies from the immune serum. This must be clearly addressed in the indirect ELISAs measuring total IgG, IgG1 and IgG2 and the western blot (Fig. 3) in the manuscript.

Ef-Tu is an elongation factor and has been identified as a vaccine candidate for A. marginale. However, it is not part of the T4SS, this should be clearly explained in the introduction, abstract and discussion.

It would be useful to use A. marginale protein from infected RBCs in a western blot to confirm that the antibody response to the recombinant proteins resulted in antibodies that can bind A. marginale proteins.

Line 209 and 219: A serum dilution of 1:10 (total IgG) and 1:20 for IgG1 and IgG2 is a very low dilution and suggests that the antibody response was potentially non-specific and weak. Please address this more fully in the manuscript.

Line 215: Please more fully describe the origin of the negative sera? Is this pre-immune serum from the calves in the study or is these sera from other known negative animals?

Line 226 and throughout the manuscript- It is difficult to put the results in context because the timeline is generally referenced based on the number of days following the first inoculation. For example, 28 days following the first inoculation is same as one week after the second inoculation. In this case is important for the reader to know that a second immunization was done and the sera were collected one week later in order to put the results in context. Please edit through the document in order to put the timeline into the context of the results.

Line 227 and Fig. 3: What was the serum dilution used for the bovine immune serum in the western blot? Please include a control on the western blot indicating that absorption was sufficient to remove the anti-his tag antibodies. Please clarify why Msp5 is used as the negative control? Does it have a His-tag? Please include the production of Msp5 in the methods. Please include pre-immune serum as a negative control.

Statistical analysis- Collapsing the data into a single data point for each parameter (PCV or PPE) for each animal results in a loss of data points. For example, PCV and PPE was determined daily for each animal, yet the statistical analysis was done with the maximum decrease in PCV for each animal or the maximum PPE for each animal, thus resulting in only one data point for each animal for each category and thus an overall loss of data and potentially statistical power. Admittedly this is unlikely to change the outcome of the analysis given the lack of differences between groups, but is still worth consideration.

Table 3: Please clarify maximum PPCV decrease. Is this the % drop in PCV? Standard deviation, as a measure of the amount of variability is more appropriate in this case than standard error.

Minor suggestions

PCV is by definition a percent and is generally represented by PCV, not PPCV. If the measurement is percent drop in PCV, just say “percent drop in PCV”.

Line 47,77- The use of the word, multiplied, is unusual in this context. Suggest, using either “produced” or “amplified” instead.

Line 102- Add GenBank accession numbers as reference for the genes.

Line 157-Add a space between dodecyl and sulfate

Fig. 1- Overall this figure needs improvement. Specifically, there needs to be more space between the coomasie and the western blot as the numbers for the ladder for the western blot overlap with the coomasie. Suggest improving figure 1 and removing the supplemental results.

Reviewer #2: Dear Editor,

Please find below the manuscript "A vaccine using Anaplasma marginale subdominant type IV secretion system recombinant proteins was not protective against a virulent challenge" PONE-D-19-26977, after complete correction.

The authors developed two formulations containing A. marginale type IV secretion system (T4SS) subdominant proteins: VirB9.1, VirB9.2, VirB10, VirB11, and Ef-Tu T4SS beind expressed in E. coli.

This proteins were absorbed on a Quil A® adjuvant (aqueous saponin adjuvant - G1) and and Montanide ISA 201 ready-to-use adjuvants for Water-in-Oil-in-Water emulsion (G2) and control groups Quil A® (G3) and ISA 201 (G4), respectively.

Recent reports describe that conserved membrane proteins that are subdominant in Anaplasma species, such as VirB9 and VirB10, are constituents of the Type 4 secretion system (T4SS) that is conserved amongst many intracellular bacteria and performs essential functions for invasion and survival in host cells (Crosby et al 2018). Therefore, here this study is important.

In addition, the work used free healthy cattle and also splentomized calves with parasitized erythrocytes from A, marginale, besides to explain the participation of constituents of the Type 4 secretion system (T4SS) on the immune response against A marginale virulent.

However I would like to request some explanations:

line 52 -54 " Such protection was correlated with induction of high titers of IgG2 opsonizing antibodies against A. marginale surface epitopes and macrophage activation mediated by CD4+ T cells [4,11,12]". To compare the works (4,11,12) and the challenge test employed. Thus, it is important to insert on the discussion section.

line 94 - 100 and 241 -267 "In silico analysis of T4SS proteins sequences". I would suggest the authors to consider the inclusion of figure or a table with the sequences.

line 172 - 182 "Immunization...I would like that authors described about the adsorption methods of the vaccine formulations employed.

It´s necessary to describe on the material and methods section.

Line 187 - 188 "Each challenge dose was adjusted to 107 parasitized erythrocytes in a 2 mL final volume and inoculated SC to the cattle, on day 105 after the first inoculation.

Why? Others reports (Santos et al 2013; Crosby et al 2018) employed 1 x 10 5 for both dominant and subdominant epitopes. it is important to insert discussion section.

line 341-349 "Despite the correlation between IgG2 and protective immunity, both vaccine formulations evaluated in this work induced a strong IgG2 response but failed to mitigate the course of infection. Other subdominant proteins from the OM, AM854 and AM936, were able to induce an immune response comparable to that induced by the whole OM, based on high levels of IgG2; however, they also failed to protect against the challenge [37]" Here, I would like that authors described about the IgG2 production by dominant epitopes, and to compare with obtained data this work (subdominant epitopes).

line 377 - 383 "Another cause of vaccine failure in this study may have been the inability of the immunogen to generate antibodies to block the parasite entry to the host cell. The most successful vaccines target highly conserved epitopes required by the pathogenic parasites for their host cell entry [42]". Here, I would like that authors described about the MSP-2 variants and the CD4(+)-T-cell epitopes that may be responsible for all or part of the A. centrale vaccine efficacy.

6. PLOS authors have the option to publish the peer review history of their article (what does this mean?). If published, this will include your full peer review and any attached files.

Reviewer #1: No

Reviewer #2: Yes: Alex Sander Rodrigues Cangussu

---

## [Author Response · Author response to Decision Letter 0]

21 Dec 2019

R. The authors need to more clearly demonstrate that they are measuring the immune response directed against the recombinant proteins and not the His tag following immunization. Though it is mentioned on line 227 that the sera were adsorbed with E. coli lysate, it is not clear that the expressed His tag was included in the E. coli lysate. Additionally, there are no controls to demonstrate that the adsorption was effective in removing the anti-his tag antibodies from the immune serum. This must be clearly addressed in the indirect ELISAs measuring total IgG, IgG1 and IgG2 and the western blot (Fig. 3) in the manuscript.

A. We made changes in the manuscript. Briefly, in the study we used three recombinant proteins (SAG1, MSA2c and MSP5) for the adsorption of antibodies anti-His-tag and for the control of the adsorption step. These proteins were expressed in E. coli with His-tag and were purified with Ni2+-NTA agarose. The sera were adsorbed with an E. coli lysate expressing SAG1 protein of Neospora caninum. Moreover, we added MSP5 of A. marginale as control to determine the presence of anti-His-tag antibodies in the WB assay (Fig 3) and we used A. marginale MSP5 and Babesia bovis MSA2c recombinant proteins to control the presence of antibodies anti-His-tag in the iELISA assay (Fig 2). These specifications were detailed in the Material and methods and Results sections. 

R. Ef-Tu is an elongation factor and has been identified as a vaccine candidate for A. marginale. However, it is not part of the T4SS, this should be clearly explained in the introduction, abstract and discussion.

A. We made the changes.

R. It would be useful to use A. marginale protein from infected RBCs in a western blot to confirm that the antibody response to the recombinant proteins resulted in antibodies that can bind A. marginale proteins.

A. We added in the WB assay (Fig 3) a control of interaction of the post-immunization antibodies with a crude antigen of A. maginale purified from the blood of a splenectomized calf with 80% of parasitemia. The crude antigen was purified following the protocol described for the Card agglutination test. 

R. Line 209 and 219: A serum dilution of 1:10 (total IgG) and 1:20 for IgG1 and IgG2 is a very low dilution and suggests that the antibody response was potentially non-specific and weak. Please address this more fully in the manuscript.

A. Although the dilution of the sera is low, we adsorbed the sera before performing ELISA and WB, with an E. coli lysate expressing SAG1. Then, we demonstrated the specificity of antibodies by the lack of reaction of sera with MSP5 and MSA2c by ELISA and WB (Fig 2 and Fig 3) and the reaction with proteins obtained from A. marginale crude antigen (Fig 3). This point was included in the discussion.

R. Line 215: Please more fully describe the origin of the negative sera? Is this pre-immune serum from the calves in the study or are these sera from other known negative animals?

A. The description of the negative sera was included in the manuscript (line 235). Pre-immune sera from cattle were used in this study. 

R. Line 226 and throughout the manuscript- It is difficult to put the results in context because the timeline is generally referenced based on the number of days following the first inoculation. For example, 28 days following the first inoculation is same as one week after the second inoculation. In this case is important for the reader to know that a second immunization was done and the sera were collected one week later in order to put the results in context. Please edit through the document in order to put the timeline into the context of the results.

A. We made the changes in the manuscript.

R. Line 227 and Fig. 3: What was the serum dilution used for the bovine immune serum in the western blot? Please include a control on the western blot indicating that absorption was sufficient to remove the anti-his tag antibodies. Please clarify why Msp5 is used as the negative control? Does it have a His-tag? Please include the production of Msp5 in the methods. Please include pre-immune serum as a negative control.

A. We included the data in the manuscript. The dilution of sera in the WB was 1:100. As control of anti-His-tag antibodies removal, MSP5-His-tag was included. MSP5 was used as a negative control because it is not part of the immunogen formulation, but contains a His-tag. The production of MSP5 was referenced (line 178). 

R. Statistical analysis- Collapsing the data into a single data point for each parameter (PCV or PPE) for each animal results in a loss of data points. For example, PCV and PPE was determined daily for each animal, yet the statistical analysis was done with the maximum decrease in PCV for each animal or the maximum PPE for each animal, thus resulting in only one data point for each animal for each category and thus an overall loss of data and potentially statistical power. Admittedly this is unlikely to change the outcome of the analysis given the lack of differences between groups, but is still worth consideration.

A. We admit the loss of data in this analysis but we chose this methodology to present the results due to the absence of differences between groups.

R. Table 3: Please clarify maximum PPCV decrease. Is this the % drop in PCV? Standard deviation, as a measure of the amount of variability is more appropriate in this case than standard error.

A. We made the changes in the manuscript. PPCV is the % drop in PCV. 

Minor suggestions

R. PCV is by definition a percent and is generally represented by PCV, not PPCV. If the measurement is percent drop in PCV, just say “percent drop in PCV”.

A. We made the change in the manuscript.

R. Line 47,77- The use of the word, multiplied, is unusual in this context. Suggest, using either “produced” or “amplified” instead.

A. We made the change in the manuscript.

R. Line 102- Add GenBank accession numbers as reference for the genes.

A. We added the access in the manuscript (lines 106 to 107).

R. Line 157-Add a space between dodecyl and sulfate

A. We made the change in the manuscript.

R. Fig. 1- Overall this figure needs improvement. Specifically, there needs to be more space between the coomasie and the western blot as the numbers for the ladder for the western blot overlap with the coomasie. Suggest improving figure 1 and removing the supplemental results.

A. We improved the figure.

R. line 52 -54 " Such protection was correlated with induction of high titers of IgG2 opsonizing antibodies against A. marginale surface epitopes and macrophage activation mediated by CD4+ T cells [4,11,12]". To compare the works (4,11,12) and the challenge test employed. Thus, it is important to insert on the discussion section.

A. The authors of reference 11 did not perform a challenge. They demonstrated that in A. marginale outer membrane-vaccinated cattle, VirB9, VirB10, and CTP are recognized by serum IgG2 and stimulate memory T-lymphocyte proliferation and gamma interferon secretion. Reference 12 is a review. 

R. line 94 - 100 and 241 -267 "In silico analysis of T4SS proteins sequences". I would suggest the authors to consider the inclusion of figure or a table with the sequences.

A. We included the sequences in the manuscript as supporting information (S1 Table).

R. line 172 - 182 "Immunization...I would like that authors described about the adsorption methods of the vaccine formulations employed.

It´s necessary to describe on the material and methods section.

A. We described the methods in the manuscript (lines 187 to 191).

R. Line 187 - 188 "Each challenge dose was adjusted to 107 parasitized erythrocytes in a 2 mL final volume and inoculated SC to the cattle, on day 105 after the first inoculation. Why? Others reports (Santos et al 2013; Crosby et al 2018) employed 1 x 10 5 for both dominant and subdominant epitopes. it is important to insert discussion section.

A. In the cited reports, the protection conferred by different antigens is evaluated in mice. In this species, the challenge dose and the challenge via are different than in cattle. Also in Crosby et al 2018, the mice are challenged with A. phagocytophilum that has tropism by a different cell (netrophils). The challenge dose depends on the species and via used. Different A. marginale challenge doses in cattle are evaluated in Abdala et al 1990.

R. line 341-349 "Despite the correlation between IgG2 and protective immunity, both vaccine formulations evaluated in this work induced a strong IgG2 response but failed to mitigate the course of infection. Other subdominant proteins from the OM, AM854 and AM936, were able to induce an immune response comparable to that induced by the whole OM, based on high levels of IgG2; however, they also failed to protect against the challenge [37]" Here, I would like that authors described about the IgG2 production by dominant epitopes, and to compare with obtained data this work (subdominant epitopes).

A. We incorporated the description in the manuscript (lines 379 to 386).

R. line 377 - 383 "Another cause of vaccine failure in this study may have been the inability of the immunogen to generate antibodies to block the parasite entry to the host cell. The most successful vaccines target highly conserved epitopes required by the pathogenic parasites for their host cell entry [42]". Here, I would like that authors described about the MSP-2 variants and the CD4(+)-T-cell epitopes that may be responsible for all or part of the A. centrale vaccine efficacy.

A. We incorporated the description in the manuscript (lines 419 to423).

---

## [Decision Letter · Decision Letter 1]

31 Jan 2020

PONE-D-19-26977R1

A vaccine using Anaplasma marginale subdominant type IV secretion system recombinant proteins was not protective against a virulent challenge

PLOS ONE

Dear miss Sarli,

Thank you for submitting your manuscript to PLOS ONE. After careful consideration, we feel that it has merit but does not fully meet PLOS ONE’s publication criteria as it currently stands. Therefore, we invite you to submit a revised version of the manuscript that addresses the points raised during the review process.

We would appreciate receiving your revised manuscript by Mar 16 2020 11:59PM. To enhance the reproducibility of your results, we recommend that if applicable you deposit your laboratory protocols in protocols.io, where a protocol can be assigned its own identifier (DOI) such that it can be cited independently in the future. For instructions see: http://journals.plos.org/plosone/s/submission-guidelines#loc-laboratory-protocols

We look forward to receiving your revised manuscript.

Kind regards,

Paulo Lee Ho, Ph.D.

Academic Editor

PLOS ONE

Reviewers' comments:

Reviewer's Responses to Questions

**Comments to the Author**

1. If the authors have adequately addressed your comments raised in a previous round of review and you feel that this manuscript is now acceptable for publication, you may indicate that here to bypass the “Comments to the Author” section, enter your conflict of interest statement in the “Confidential to Editor” section, and submit your "Accept" recommendation.

Reviewer #1: (No Response)

Reviewer #2: All comments have been addressed

2. Is the manuscript technically sound, and do the data support the conclusions?

Reviewer #1: Yes

Reviewer #2: Yes

3. Has the statistical analysis been performed appropriately and rigorously? 

Reviewer #1: Yes

Reviewer #2: Yes

4. Have the authors made all data underlying the findings in their manuscript fully available?

Reviewer #1: Yes

Reviewer #2: Yes

5. Is the manuscript presented in an intelligible fashion and written in standard English?

Reviewer #1: Yes

Reviewer #2: Yes

6. Review Comments to the Author

Reviewer #1: The authors have addressed all concerns and suggestions from this reviewer. We appreciate their efforts. Below are a few more minor suggestions, mostly regarding word choice:

Line 15: Suggest replacing “multiplied” with “produced” or “amplified”.

Line 16: suggest replacing “vector” with “carry” or “transmit”

Line 18: Replace “works” with “work”

Line 23 and 26: Cattle from G1 and G2 were inoculated with a mixture of 50 ug of each recombinant protein with QuilA and Montanide adjuvants. Edit as follows: Cattle from G1 and G2 were inoculated with a mixture of 50 ug of each recombinant protein with QuilA or Montanide adjuvants, respectively.

Line 26: “Cattle received four doses…”. I suggest: Cattle received four immunizations…”

Line 28, line 245, 294, 298, 300, 311, 314, 319 and throughout the manuscript, for clarity, I suggest using immunization instead of inoculation when referring to administration of the antigens. This more specific language helps differentiate between immunization and challenge.

Line 289: “(B) Western blot revealed with…” suggest “(B) Western blot using anti-His-tag….”

Line 312: Replace Babesia bovis with B. bovis

Line 315: Replace Anaplasma marginale with A. marginale

Reviewer #2: I'm satisfied with the authors responses and recommend the publication of the work. However, I suggest a minor revision as described in the attached. Grateful for the opportunity to evaluate this work. The work it is important to elucide about the subdominant epitopes participation of Anaplasma marginale and the immunization of bovines.

7. PLOS authors have the option to publish the peer review history of their article (what does this mean?). If published, this will include your full peer review and any attached files.

Reviewer #1: No

Reviewer #2: Yes: Alex Sander Rodrigues Cangussu

---

## [Author Response · Author response to Decision Letter 1]

31 Jan 2020

Reviewer #1

Line 15: Suggest replacing “multiplied” with “produced” or “amplified”.

We made the change in the manuscript.

Line 16: suggest replacing “vector” with “carry” or “transmit”

We made the change in the manuscript.

Line 18: Replace “works” with “work”

We made the change in the manuscript.

Line 23 and 26: Cattle from G1 and G2 were inoculated with a mixture of 50 ug of each recombinant protein with QuilA and Montanide adjuvants. Edit as follows: Cattle from G1 and G2 were inoculated with a mixture of 50 ug of each recombinant protein with QuilA or Montanide adjuvants, respectively.

We made the change in the manuscript.

Line 26: “Cattle received four doses…”. I suggest: Cattle received four immunizations…”

We made the change in the manuscript.

Line 28, line 245, 294, 298, 300, 311, 314, 319 and throughout the manuscript, for clarity, I suggest using immunization instead of inoculation when referring to administration of the antigens. This more specific language helps differentiate between immunization and challenge.

We made the changes in the manuscript.

Line 289: “(B) Western blot revealed with…” suggest “(B) Western blot using anti-His-tag….”

We made the change in the manuscript.

Line 312: Replace Babesia bovis with B. bovis

We made the change in the manuscript.

Line 315: Replace Anaplasma marginale with A. marginale

We made the change in the manuscript.

Reviewer #2 

Line 52 – 54 Here, I would like that authors excluded "protection", because mentioned authors did not perform the challenge test. If possible, to replace the reference 12 by : Linkage between Anaplasma marginale Outer Membrane Proteins Enhances Immunogenicity but Is Not Required for Protection from Challenge. Clin Vaccine Immunol. 2013 May; 20(5): 651–656. Noh et al 2013

We replace the reference 12 and exclude the reference 11 because it has not challenge test. (line 55).

Line 202-203 Noh et al 2013 (Linkage between Anaplasma marginale Outer Membrane Proteins Enhances Immunogenicity but Is Not Required for Protection from Challenge) it also describes about the challenge test of bovine (Holstein steers) and also can be used to discute the work.

The challenge dose, the challenge via (subcutaneous or intravenous), the challenge method (infected blood or infected ticks) and the challenge time post-immunization are different in the bibliography. However, in all works A. marginale OM induced protection (Tebele et al 1991, Brown et al 1998, Ducken et al 2015, Albarrak et al 2012, Noh et al 2013). For this reason we decide not include the point in the discussion.

---

## [Editor Report · Decision Letter 2]

4 Feb 2020

A vaccine using Anaplasma marginale subdominant type IV secretion system recombinant proteins was not protective against a virulent challenge

PONE-D-19-26977R2

Dear Dr. Sarli,

We are pleased to inform you that your manuscript has been judged scientifically suitable for publication and will be formally accepted for publication once it complies with all outstanding technical requirements.

With kind regards,

Paulo Lee Ho, Ph.D.

Academic Editor

PLOS ONE
---

## [Editor Report · Acceptance letter]

6 Feb 2020

PONE-D-19-26977R2 

A vaccine using *Anaplasma marginale* subdominant type IV secretion system recombinant proteins was not protective against a virulent challenge 

Dear Dr. Sarli:

I am pleased to inform you that your manuscript has been deemed suitable for publication in PLOS ONE. Congratulations! Your manuscript is now with our production department. 

With kind regards,

on behalf of

Dr. Paulo Lee Ho 

Academic Editor

PLOS ONE